# Method to evaluate large wood behavior in terms of convection equation associated with sediment erosion and deposition

Daisuke HARADA[1], Shinji EGASHIRA[1]

[1]International Centre for Water Hazard and Risk Management (ICHARM), Public Works Research Institute (PWRI), 1-6, Minamihara, Tsukuba, Ibaraki, 305-8516, Japan

*Correspondence to*: Daisuke HARADA (d-harada55@pwri.go.jp)

**Abstract.** Recent flood hazards occurring in mountainous areas are often characterized by numerous amounts of sediment and large wood supplied from upstream, which often exacerbate flood disasters in downstream areas. This paper proposes a method for describing large wood behavior in terms of the convection and the storage equations, together with the governing equations for describing flood flows and channel changes associated with active sediment erosion and deposition. The proposed method is tested for its validity by simulating the phenomena occurring in an open channel with an erodible bed, and flood flow with numerous amounts of sediment and large wood in the Akatani River flood disaster event. As a result of calculations reproducing the open channel experiment, the applicability of the method is indicated as the percentage of wood pieces captured in the sediment deposition areas in the channel is within the range of the experimental results. The results of 2-D flood flow calculations with sediment and large wood in the Akatani river flood disaster suggested that large wood deposition is reproduced where bed deformation is well reproduced. Overall, since the proposed method makes it possible to simulate the behavior of a numerous number of large wood, it can be applied to the management of hazards in mountainous rivers such as the Akatani River.

## 1 Introduction

On July 5, 2017, extremely heavy rains hit northern Kyusyu, Japan, causing landslides and debris flows at numerous locations in mountainous areas. In Japan, flood disasters with a large amount of sediment deposition are becoming increasingly apparent in mountainous areas or steep slope areas. This is caused by sediment and large wood pieces transported by landslides and debris flows associated with heavy rainfall and deposited at small valley outlets, transported by flood waters, and inundated with flood waters where the sediment transport capacity is rapidly reduced.

The Akatani River, with a basin area of approximately 20 km$^2$, was particularly severely damaged in this disaster (Chen et al., 2018; Harada and Egashira, 2018). The photo on the left side of Figure 1, taken immediately after the disaster, shows that the sediment and large wood carried by the debris flow were accumulated at the valley outlet. According to the results of the laser profiler survey, approximately 2.9 million m$^3$ of sediment was produced by landslides and debris flows in the Akatani River basin. Multiplying the area of the landslide and debris flow area by 549 m$^3$/ha of timber volume per unit area, it is estimated

that approximately 39,000 m$^3$ of large wood was produced (MLIT, 2017). This means that assuming the standing tree density as one tree in 2m$^2$, approximately 19,500 pieces of large wood were produced and carried to their deposited regions during the event in the Akatani river basin. Such sediment and large wood produced by landslides and debris flows are deposited in areas where the slope is about 4 to 6 degrees, often forming a debris flow fan at the exit of valleys, as shown in the photo on the left side of Figure 1. When the deposited sediment and large wood are eroded by the water flow, part of them is transported with

the flood flow through the channel network, and once such large wood is trapped on bridges and other structures, it disrupts the flood flow. The photo on the left side of Figure 1, taken immediately after the disaster, shows that considerable amounts of large wood were accumulated at around the bridge. Recently, these types of flood hazards have been reported to occur in many places of steep, forested areas (Lucía et al., 2015; Lucía et al., 2018; Steeb et al., 2017; Comiti et al., 2016; Harada et al., 2019). As reported in those studies, large wood pieces often become a major contributing factor that exacerbates flood

disasters. Hence, it is highly important to develop a method to evaluate their behavior in a flood flow.

Research on large wood in rivers has been conducted, and some studies have proposed simulation models for their behavior (Swanson et al., 2021; Ruiz-Villanueva et al., 2019). Nakagawa et al. (1994) proposed a numerical simulation model to compute the behavior of individual wood pieces in a two-dimensional flow field by calculating the transportation of wood using a combination of translocation and rotation. This method was applied and verified in several laboratory experiments

(Shrestha et al., 2009; Shrestha et al., 2012). Gotoh (2001) proposed a method to track the motion of driftwood based on the Lagrange particle method (MPS method) by treating wood pieces as rigid bodies. Shimizu et al. (2006) developed a model composed of two types of analyses: the Eulerian analysis of fluid motion, using depth-averaged flow analysis, and the Lagrange analysis of driftwood motion, using the extended distinct element method. Ruiz-Villanueva et al. (2014) proposed a method to compute the transport of individual large wood pieces in a two-dimensional flow field, including physical modeling of the

wood recruitment process to the flood flow. Kimura et al. (2021) developed models to compute large wood motion in a three-dimensional flow field.

In addition to those proposed methods for the behaviour of wood pieces in a flow field, some research proposed methods to analyse large wood production processes from a watershed. Benda and Sias (2003) proposed stochastic methods to predict long term wood budgets in watersheds. Mazzorana et al. (2009) proposed methods to estimate large wood production processes

in mountainous streams by a GIS-based index. Mazzorana et al. (2011) proposed a model for wood entrainment and deposition processes based on empirical methods and the transportation of wooden materials in the flow field. Komori et al. (2021) proposed a model to evaluate large wood export at a watershed scale.

As suggested in the previous descriptions, the disasters that have occurred in mountainous areas in recent years can be divided into two categories: those caused by landslides and debris flows and their direct inundation, and those caused by floods in

areas where the flood and sediment transport capacity of the succeeding flood flows is rapidly reduced. The latter is generally accompanied by large amounts of sediment and large wood pieces, as flood flows erode and transport the sediment carried by landslides and debris flows. Therefore, in order to understand the characteristics of disasters in mountainous areas, it is not only important to elucidate the erosion and transport mechanisms of sediments and large wood associated with landslides and

debris flows, but also essential to understand the erosion and transport mechanisms of sediments and large wood carried by

flood flows. This study focuses on the latter topic.

Previous Lagrangian modeling attempts, such as those proposed by Shimizu et al. (2006) and Ruiz-Villanueva et al. (2014), allow for detailed evaluation of large wood behaviour by analyzing individual wood pieces in the flow. However, it is difficult to apply such Lagrangian methods to flood disasters such as the one in the Akatani River, which produced as many as 19,500 large wood pieces with active sediment erosion and deposition. Therefore, to treat such a large wood behaviour with active

sediment transport, this study proposes a method to describe the behavior of large wood based on a convection equation and a storage equation considering sediment erosion and deposition. Then, to investigate the validity of the proposed method in terms of erosion and deposition rates of large wood, a series of calculations are performed to reproduce the experimental results in a straight open channel. In addition, we demonstrate the application to the 2017 flood disaster in northern Kyusyu by calculating flood flows using a 2-D depth-averaged flow model with sediment and large wood. Although the reproducibility

of the field application may not be perfect due to its complexity, we present it here since our target is to use the proposed method to evaluate and predict possible sediment and large-wood hazards and mitigate them in rivers.

## 2 Methods

To evaluate flood flows and their flooding process influenced by channel changes and large wood traps, we employ depth

averaged 2-D governing equations consisting of mass and momentum conservation equations. The equations are expressed in the Cartesian coordinate system as follows:

$$\frac{\partial h}{\partial t} + \frac{\partial uh}{\partial x} + \frac{\partial vh}{\partial y} = 0 \tag{1}$$

$$\frac{\partial hu}{\partial t} + \frac{\partial huu}{\partial x} + \frac{\partial huv}{\partial y} = -gh\frac{\partial(h+z_b)}{\partial x} - \frac{\tau_x}{\rho} + \frac{1}{\rho}\left(\frac{\partial h\sigma_{xx}}{\partial x} + \frac{\partial h\tau_{yx}}{\partial y}\right) \tag{2}$$

$$\frac{\partial hv}{\partial t} + \frac{\partial huv}{\partial x} + \frac{\partial hvv}{\partial y} = -gh\frac{\partial(h+z_b)}{\partial y} - \frac{\tau_y}{\rho} + \frac{1}{\rho}\left(\frac{\partial h\tau_{xy}}{\partial x} + \frac{\partial h\sigma_{yy}}{\partial y}\right) \tag{3}$$

where $x$ and $y$ are the coordinates in the major flow direction and normal to the flow direction, respectively; $t$ is the time; $h$ is the flow depth; $u$ and $v$ are the $x$ and $y$ components of the depth-averaged velocity; $g$ is the acceleration due to gravity; $\rho$ is the mass density of water; $\sigma_{xx}$, $\sigma_{yy}$, $\tau_{xy}$, and $\tau_{yx}$, are the depth-averaged Reynolds stresses; $z_b$ is the bed elevation; $\tau_x$ and $\tau_y$ are

the $x$ and $y$ components of the bed shear stress, respectively.

Equations (1) to (3) are transformed into a general coordinate system (Shimizu and Itakura, 1991). The equations are numerically calculated by the cubic interpolated pseudo-particle (CIP) method (e.g., Yabe et al., 1991, Jang and Shimizu, 2005).

To evaluate the processes of bed deformation caused by bedload, suspended load, and suspended sediment transport in the 2-D flow field where the sediment sorting takes place actively, the following equatoins are employed:

$$\frac{\partial z_b}{\partial t} + \frac{1}{1-\lambda} \sum_i \left( \frac{\partial q_{bix}}{\partial x} + \frac{\partial q_{biy}}{\partial y} + E_i - D_i \right) = 0 \tag{4}$$

$$\frac{\partial c_i h}{\partial t} + \frac{\partial u c_i h}{\partial x} + \frac{\partial v c_i h}{\partial y} = E_i - D_i \tag{5}$$

where $\lambda$ is the porosity of the bed sediment; $q_{bix}$ and $q_{biy}$ are the $x$ and $y$ components of the bedload transport rate for grain size $d_i$, respectively, $E_i$ and $D_i$ are the erosion and deposition rates of the suspended sediment for grain size $d_i$, respectively, $c_i$ is the depth-averaged suspended sediment concentration for grain size $d_i$.

The bedload transport rate is estimated using a formula developed by Egashira et al. (1997), in which the constitutive relations of a water-sediment mixture flow are applied to the bedload layer.

$$q_{b*i} = \frac{4}{15} \frac{K_1^{\,2} K_2}{\sqrt{f_d + f_f}} \tau_{*i}^{\,5/2} p_i \tag{6}$$

where $q_{b*i}$ is the non-dimensional bedload transport rate for grain size $d_i$; $\tau_{*i}$ is the non-dimensional tractive force for grain size $d_i$; $p_i$ is the content ratio for grain size $d_i$; the other parameters, $K_1$, $K_2$, $f_d$, and $f_f$, are specified based on Egashira et al. (1997).

$$K_1 = \frac{1}{\cos \theta} \frac{1}{\tan \phi_s - \tan \theta} \tag{7}$$

$$K_2 = \frac{1}{\bar{c}_s} \left[ 1 - \frac{h_s}{h} \right]^{1/2} \tag{8}$$

$$f_d = k_d (1 - e^2) \frac{\sigma}{\rho} \bar{c}_s^{\,1/3} \tag{9}$$

$$f_f = k_f (1 - \bar{c}_s)^{5/3} \bar{c}_s^{\,-2/3} \tag{10}$$

where $\theta$ is the local slope; $\phi_s$ is the angle of repose; $\bar{c}_s$ is the sediment concentration on the bedload layer; $h$ is the water depth; $\sigma$ is the density of soil particle; $\rho$ is the density of water; $e$ is the restitution that determines the energy loss in binary collision; $k_d$=0.0828; $k_f$=0.16. Indeed, most of the values in equations (7) to (10) are regarded as constant in the flow field described by equations (1) to (3), thus assuming, $\phi_s = 34°$, $\bar{c}_s = 0.25$, $e = 0.85$, and $h_s/h \ll 1$, the value of the term $4/15\, K_1^{\,2} K_2/\sqrt{f_d + f_f}$ in equation (6) is approximately 4.35.

In equation (8), $h_s$ is the thickness of the bedload layer, which is described as follows (Egashira et al., 1997):

$$\frac{h_s}{h} = \frac{1}{(\frac{\sigma}{\rho} - 1) \bar{c}_s} \frac{\tan \theta}{\tan \phi_s - \tan \theta} \tag{11}$$

The grain size distribution of bed materials is evaluated based on the concept of the bedload layer, the transition layer, and the deposition layer, which was developed by Luu et al. (2006), assuming that the mass of each material is preserved.

Erosion rate $E_i$ of suspended sediment in equations (4) and (5) are evaluated using the following equations proposed by Harada et al. (2022);

$$E_i = p_i W_e \bar{c}_s \tag{12}$$

$$\frac{W_e}{U} = \frac{K}{R_{i*}} \tag{13a}$$

$$R_{i*} = (\sigma/\rho - 1)\bar{c}_s gh/\sqrt{u^2 + v^2}^{\,2} \tag{13b}$$

where $W_e$ is the entrainment velocity at the boundary between the upper water layer and the bedload layer, $R_{i*}$ is the overall Richardson number, $c$ is the depth-averaged suspended sediment concentration, and $K = 1.5 \times 10^{-3}$ (Egashira and Ashida, 1980). Note that Equations (6) and (12) are used taking into account the importance of sediment sorting when evaluating the flow field with active channel changes and bed deformations.

To analyze considerable amounts of large wood in the flood flow, we assume that the wood pieces behave as neutral buoyant particles that do not affect the flow structure, as this assumption allows the introduction of the convection equation. This means that large wood pieces are transported as fine particles with the flow. In addition, the exchange of these particles to large wood pieces, such as the wood deposition from the water to the riverbed, the wood recruitment from the riverbed to the water, and the wood entrapment on structures, should also be considered. For this purpose, the convection equation is coupled with the storage equation for large wood in the channel bed, assuming that wood erosion and deposition are proportional to sediment erosion and deposition, and that wood accumulation occurs at artificial structures such as bridges. This concept is consistent with the idea that when sediment is deposited on the riverbed, water is also deposited as pore water within the void of the sediment, and when sediment is eroded, pore water is also taken into the water, and this eroded sediment and pore water contains large wood, as the wood particles are treated in terms of wood concentration.

Based on these assumptions, the behavior of large wood in the flood flow is expressed using the following equations:

$\partial z_b / \partial t > 0$:

$$\frac{\partial c_{drf} h}{\partial t} + \frac{\partial c_{drf} uh}{\partial x} + \frac{\partial c_{drf} vh}{\partial y} = -c_* \frac{\partial z_b}{\partial t} c_{drf} r(t,x,y) - v_n c_{drf} p_b \delta(x - x_i, y - y_i) \tag{14}$$

$$\frac{\partial S}{\partial t} = \frac{\partial z_b}{\partial t} c_{drf} r(t,x,y) + v_n c_{drf} p_b \delta(x - x_i, y - y_i) \tag{15}$$

$\partial z_b / \partial t < 0$:

$$\frac{\partial c_{drf} h}{\partial t} + \frac{\partial c_{drf} uh}{\partial x} + \frac{\partial c_{drf} vh}{\partial y} = -c_* \frac{\partial z_b}{\partial t} \frac{S}{D} r(t,x,y) - v_n c_{drf} p_b \delta(x - x_i, y - y_i) \tag{16}$$

$$\frac{\partial S}{\partial t} = \frac{\partial z_b}{\partial t} \frac{S}{D} r(t,x,y) + v_n c_{drf} p_b \delta(x - x_i, y - y_i) \tag{17}$$

where $c_{drf}$ is the depth-averaged large wood concentration; $c_*$ is the sediment concentration of the stationary bed; $v_n$ is the inward velocity normal to the structure area such as the bridge; $D$ is the depth of the standing tree's root; $S$ is the stored volume

of large wood in a unit area of the ground or the riverbed per unit area; when the volume of a piece of wood is $V$, $S$ and the number of wood ($N$) in a certain area ($A$) is converted by $S = VN/A$.

Equations (14) and (16) are the convection equations for large wood transport with the water flow, and equations (15) and (17) are the storage equations of large wood stored on the bed. The first term of the right-hand side in equations (14) to (17) represents the large wood exchange between the water flow and the channel bed. Since we assume that the erosion and deposition of large wood take place in proportion to sediment erosion and deposition, the term in equations (14) and (15) represents the wood deposition from the water to the bed, and the term in equations (16) and (17) represents the large wood recruitment from the bed to the water. Figure 2 (a) shows the concept of these processes.

In the case of sediment deposition, $c_* \partial z_b/\partial t$ corresponds to the sediment deposition rate per unit time in unit area. We assume that large wood pieces within the height range are entrained into the riverbed; thus, the amount of $c_* \partial z_b/\partial t\, c_{drf}$ is stored in the riverbed. In the case of sediment erosion, as shown in Figure 2 (b), when the bed erosion reaches root depth $D$, all wood storage $S$ is recruited to the water. Therefore, $S\, c_* \partial z_b/\partial t\, /D$ corresponds to the large wood recruitment from the bed to the water per unit time.

Large wood recruitment does not occur at depths shallower than a certain water depth, and large wood deposition does not occur at depths deeper than a certain water depth. Function $r(t, x, y)$ in equations (14) to (17) is introduced to describe these cases and set as shown in Figure 3 in the present research.

In equations (14) and (16), the last term defines large wood capture at the structures. Dirac's δ-function is employed to evaluate the capturing of large wood at structures such as bridges by defining the locations of these types of structures as $((x,\ y) = (x_i, y_i))$ and setting Dirac's δ-function for the structures as δ = 1 and for other places as δ = 0. $p_b$ denotes the probability that large wood is captured at structures, ranging from 0 to 1.

When large wood is present in the bed, the sediment transport rate may be reduced due to the cover or shielding effect of the large wood. Therefore, in the present study, the bedload transport rate in equation (6) and the erosion rate of suspended sediment in equation (12) are reduced at the rate of the shielding effect.

At a point where a structure, such as a bridge, exists, e.g., $((x,\ y) = (x_i, y_i))$, the water decreases in the cross section where the velocities across the cell, i.e., $u\Delta x$ and $v\Delta y$, are reduced by $u\alpha\Delta x$ and $v\alpha\Delta y$, in which α is described as follows:

$$\alpha = \frac{S}{h}/(1 - c_{*drf}) \tag{18}$$

where $c_{*drf}$ is the wood concentration of a stationary layer composed of the deposited large wood only.

## 3 Tests of the erosion and deposition terms of large wood

### 3.1 Outline

In the proposed method, we assume two major factors that affect calculation results: the exchange of the large wood particles between the water and the bed and wood entrapment on structures. In order to investigate the validity of these factors, we

compare the results of previous flume experiments and the calculations performed to reproduce the flume experiments. In the flume experiment conducted by Itoh et al. (2010), a sediment bed was prepared in the flume, and water and wood pieces were constantly supplied from the upstream end to investigate the deposition rate of the woood pieces in the flume. The original purpose of the experiment was to investigate the effect of the specific gravity of wood pieces on their deposition rate, and since the present method assumes that the erosion and deposition of large wood take place in proportion to sediment erosion and deposition, this experiment is suitable for investigating the validity of the two factors.

In some cases of the experiment where the flume width and the log length were the same, once a log was deposited in such a way as to block the cross-section of the flume, other logs were caught one by one with that first log, resulting in the log jam phenomenon. The log jam formed in the flume in run 16 is shown in Figure 4.1. This log jam does not occur in the calculation without a trigger because the present method treats logs as the depth-averaged concentration of large wood. Therefore, in some cases of the calculation, obstacles that trap wood pieces, i.e., the locations where $\delta=1$ in equations (14) to (17) with $p_b = 1$, are set in the channel-crossing direction as a trigger so that wood is trapped there.

### 3.2 Experiments

In the flume experiment conducted by Itoh et al. (2010), a straight open channel with a length of 10 m, a width of 20 cm, and a slope of 0.045 was employed. The experimental cases and the results are shown in Table 1, where runs 1 to 8 were conducted without sediment; thus, these cases are not included in this study. In runs 9 to 15, uniform sediment with a grain size of 18.3mm in $d_{60}$ was continuously supplied from the upstream end, so that the sediment equilibrium condition was achieved during the experiment. The flow discharge in the experiment was kept constant at 1.42 (l/s), and pieces of wood with a length of 20 cm, equal to the width of the flume, and a diameter of 0.61 cm were randomly supplied from the upstream end for 60 seconds. Throughout the experiment, the log entrapment rate, i.e., the ratio of logs trapped in the flume to logs supplied, was measured. As shown in Table 1, the wood models were made of polyethylene with a specific gravity of 0.95, polymethyl methacrylate with a specific gravity of 1.20, or a 1:1 mixture of the two. The wood supply rate varied from case to case, and basically, the higher the supply rate, the higher the log entrapment rate due to the higher probability of log jam formation. In the cases where a log jam was not formed, the log entrapment rate ranged from 0 to 3%, and the logs were trapped where sediment deposition occurred along with the bed forms. In the cases where a log jam was formed, the log entrapment rate ranged up to 77.2%. In cases where two log jams were formed, more logs tended to be trapped.

### 3.3 Numerical simulations

A depth-averaged 2-D flood flow model with sediment transport and bed deformation, iRIC-Nays2DH (Shimizu et al., 2019), which is partially modified by the authors, is employed to reproduce the flume experiments. The computational grid size is 1 cm square, and the other computational conditions are the same as in the experiment conditions shown in Chapter 3.2. Upstream wood concentration $c_{drf}$ is set to 0.41%, which corresponds to the wood supply rate of 1 (log/s) in the experiment. To induce bed deformations in the calculation, small perturbations with a grain size scale are randomly applied to the initial

bed. Therefore, before the logs are supplied from the upstream end, the channel is sufficiently close to equilibrium with sand bars that have formed in the bed. The computational cases and the results of the log entrapment rates are shown in Table 2. In cases 4 to 7, obstacles that trap wood pieces are set, as shown in Figure 5, to induce log jams.

In the computation case 1, sand bars are formed, as shown in Figure 6. The dotted line in the top figure shows the front line of the sand bar just before the wood supply begins, and the solid line shows the front line of the sand bar after 60 seconds, meaning

that sediment deposition occurs between these two lines due to the sand bar migration. As shown in Figure 6 below, wood is deposited where sediment is deposited because the present method assumes that wood deposition occurs in response to sediment deposition. Function $r(t, x, y)$ in equations (14) through (17) is expected to affect the wood deposition. As shown in Figure 3, $r(t, x, y)$ is set to 0 when the ratio of water depth to wood diameter is greater than 2. This is because we assume that if there are branches in the wood, the wood will be deposited on the riverbed even if the ratio of water depth to wood

diameter is greater than 1. To investigate the effect of this function, $r(t, x, y)$ is changed as shown in cases 2 and 3. According to the results of cases 2 and 3 in Table 2, the log entrapment rate is about 73% greater in case 2 than the rate in case 1.

In cases 4 to 7, the obstacle lines are set. For example, the log entrapment rate is 16.7% in case 4, and most of the logs are caught at the obstacle. Since the log entrapment rate is proportional to the inward velocity normal to structure $v_n$, some logs are trapped at the obstacle, and the remaining logs flow downstream. In the experiment, not only one log but several logs are

deposited in the same place, which causes the formation of a jam. Case 5 is designed for this situation, with two rows of grids aligned across the channel as obstacles. As a result, 40% more logs are captured in case 5, compared to case 4. Case 6 is designed to form two log jams, with two rows of obstacles placed in separate locations. As a result, about twice as much wood is captured in case 6 as in case 4. In case 7, two rows of obstacles are set in each jam, as in case 5. As a result, about 47% more logs are captured in case 7 than in case 6.

## 4 Application to the Akatani river flood disaster

### 4.1 Target areas and hazard characteristics

The present method has been proposed to simulate the behavior of a large number of large wood pieces in a flood flow where active sediment transport and channel change occur. Therefore, we apply the method to the actual flood disaster in the Akatani River in 2017 to investigate the applicability and characteristics of the proposed method. The Akatani River basin is located

on the right-bank side of the Chikugo River, where a large amount of sediment and large wood were produced in the 2017 flood disaster. The drainage area and the stream length of the Akatani River are approximately 20 km$^2$ and 8 km, respectively. According to Nagumo & Egashira (2019), 639 houses or buildings in the basin were damaged during the event. Figure 7 shows the Akatani River basin with debris flows and flood marks identified from aerial photos. This shows that numerous landslides and debris flows occurred in the mountainous areas, which increased damage to the downstream areas. Figure 8 compares

aerial photos taken before and after the disaster. Although it is difficult to identify the river channel in the photo before the

event because of its very narrow width, the photo after the event clearly shows sediment widely spreading over the valley bottom, indicating highly active sediment transport and deposition during the event.

Figure 9 shows the sediment size distribution observed immediately after the flood event. Longitudinal sediment sorting is clearly observed, exhibiting a tendency for the sediment size to become finer downstream. Moreover, although the average bed slope of the entire Akatani River channel is approximately 1/70, the grain size of the deposited sediment is quite fine, which indicates that a large amount of fine sediment was supplied from the upstream area during the event.

During the flood event, numerous amounts of large wood were supplied to the channels. Figure 10 shows the distribution of deposited wood length, which is identified from aerial photos taken just after the event. Three areas along the Akatani River channel and one valley in the basin shown in Figure 7 are selected to investigate the length of each wood piece, for the resolution of the aerial photos taken for these areas is sufficient for this purpose. According to Figure 10, the distribution of wood length decreases from the inside of the valley to its outlet. A part of the large wood deposited in the outlet of the valley may have been transported to downstream due to the flood flow and deposited along the Akatani River channel.

## 4.2 Upstream boundary conditions

To conduct a 2-D depth-averaged analysis under the conditions such as the Akatani River disaster, it is necessary to evaluate the amount of sediment and large wood inflow from the basin at the upstream boundary of the 2-D analysis area. In this study, we obtained the upstream boundary condition by an integrated method to simulate rainfall-runoff, landslide and debris flow, and sediment and large wood transport in the river channel to obtain a time series of sediment and large wood discharged from the basin.

The Rainfall-runoff-inundation (RRI) model, developed by Sayama et al., (2012), is prepared for the entire basin. The model deals with slopes and river channels separately. The flow on the slope grid cells is calculated with the 2D diffusive wave model, while the channel flow is calculated with the 1D diffusive wave model. On the slope grid cells, the slope stability analysis and debris flow computations (Yamazaki et al., 2016, Yamazaki and Egashira, 2019) are conducted. The occurrence of landslides is determined using the balanced equation of a force's action on an infinite slope. When the landslide occurrence is detected, the surface soil in the cell is transported from the point of origin to the location where the deposition occurs, using the equation of a mass system. Along with the sediment transport due to the landslide and debris flow, the standing woods there is also recruited to the debris flow, and transported following the one-dimensional notation of equations (14) to (17).

When the debris flow reaches the river channel grid cells, sediment and large wood are treated as sediment and large wood supply to the river channel. In the river channel grid cells, sediment and large wood transport is evaluated with the methods proposed by Egashira and Matsuki (2000), in which a section that includes the upstream confluence and excludes the downstream confluence point is designated as the unit channel, and the sediment and large wood runoff for the entire basin is predicted by allocating the unit channels in series and parallel. As for the large wood transport in the channel network, the behavior of the large wood in the unit channel follows the one-dimensional forms of equations (14)-(17).

These models are applied to the Akatani River event in 2017 to estimate the time series of water, sediment, and large wood discharged from the basin. JMA analytical rainfall data is given as the rainfall data for the model. Since there is no hydrological record in the Akatani River basin, the model parameters were calibrated using the data from the Kagetsu River basin, which is located east of the study basin, and the parameters are applied to estimate the flow discharge in the Akatani River basin. As a result, we estimate the peak discharge as approximately 340 (m$^3$/s) at the 3.5 km point; the location corresponds to the upstream boundary of the 2-D flood flow computation, which is close to those of Shakti et al. (2018) (400 m$^3$/s) and the Ministry's reports (MLIT, 2017) (400 m$^3$/s).

Parameters employed for the rainfall-runoff and landslide calculations are shown in Table 3. As for the landslide and debris flow, model parameters are validated so that sediment transport locations and its total areas are close to those of Figure 7 and the Ministry's reports (MLIT, 2017). The initial conditions of sediment size distribution in the river channel are shown by the red dotted line in Figure 9. As for the large wood runoff computation from the basin, referring to our surveys and Kubota (2019), the density of standing trees is set as 0.06 (m$^3$/m$^2$), assuming that the average diameter of a standing tree is 15 cm, the length 11.2 m, and the density per standing tree 2 m$^2$.

The upstream boundary conditions obtained using this method are shown in Figure 11. The Figure shows the temporal variation of the basin scale computational results for flood water, suspended sediment, and large wood discharge at the 3.5 km point; the location corresponds to the upstream boundary of the 2-D flood flow computation. According to the figure, suspended sediment and large wood discharge are concentrated before the flow discharge peak comes. Note that although it is difficult to validate these boundary conditions due to the complexity of the field and the lack of data, the results appear reasonable based on the information we have and sufficient for the present purpose, which is to investigate the applicability and characteristics of the proposed method.

### 4.3 Computational conditions for the 2-D flood flow with sediment and large wood behavior

The computation area is approximately 3.5 km long, as shown in Figure 7. The average bed slope of the computational domain within the 3.5km is approximately 1/120. For the computation, iRIC-Nays2DH (Shimizu et al., 2019), which was partially modified by the authors, is employed. As the initial topography, DEM data measured by an aerial laser survey before the flooding are used. The roughness coefficient is set as 0.03 for the entire computation domain, which was determined so that the flood marks would generally match the computation results. The initial sediment size distribution, indicated by the red dotted line in Figure 9, is given within the 3.5km reach. The grid size is 5m by 5m. No large wood deposition is set, i.e., S is set to 0 in Eqs. (14)-(17) in the entire calculation domain.

Seven bridges inside the domain are set as obstacles, and $\delta = 1$ in equations (14)-(17) at these locations. The large wood capture rate $p_b$ at bridges can take values between 0 and 1, but in this study, $p_b$ is uniformly set to 1. In this computation, when large wood accumulation takes place on bridges, the cross-section area of the flow in the grid is reduced, which in turn affects the flow conditions around the bridges. Calculations are performed for the three cases shown in Table 4 to compare the differences in results depending on the presence of sediment and large wood. Case 1 is the flow computation only without

sediment and large wood, Case 2 is the flow with sediment without large wood, and Case 3 is the flow with sediment abd large wood.

## 4.4 Computation results

Figure 12 compares Cases 1, 2, and 3, and shows a water depth contour map at the peak discharge time, which shows the area
between 1.2 km and 2.5 km from the downstream end of the computational domain. For example, the area circled by the solid white line in Figure 9 shows that a wider area is inundated in Case 3 than in Case 1, which is closer to the actual inundated area.

Figure 13 compares the difference between the ground elevation measured by an aerial laser survey before and after the flooding (left figure) and the difference between the beginning and the end of the calculation for the ground elevation in Case
3 (right figure). Although the calculation results show a little excessive sediment deposition upstream of the 1.5 km point, the two trends are generally consistent in that more than 2m sediment deposition takes place in the river channel and that sediment is deposited several tens of centimeters to 1m thick in the areas where inundation occurred.

Figure 14 compares the observed number of large-wood pieces deposited in a 25-m square area with the number of computed pieces deposited at the end of the calculation. The number of observed large wood pieces deposited within a 25-m square area
is determined from aerial photographs taken immediately after the flood event. The computed results, i.e., S in equations (14)-(17), are converted to the number of large wood pieces by assuming that the diameter and length of a piece of wood are 20 cm and 7 m, respectively, referring to Figure 10 and Kubota (2019). It should be noted that some of the deposited large wood pieces are not distinguished because they are buried.

Figure 15 shows the water level in the river channel and the riverbed height at the peak flow in three different cases and
compares the results with the trace water level. In Case 3, the accumulation of large wood near the bridge obstructs the river channel flow; thus, the water level rises markedly upstream of the bridge. Comparing the trace water level and the calculated water level around the 1.5 km point, the water levels in Cases 1 and 2 are about 1m lower than the trace water level, and the water level in in Case 3 is partly due to the large wood capture rate $p_b$ at the bridge is uniformly set to 1, but at least the water level is evaluated lower in Cases 1 and 2, where large wood is not considered. In Case 3, the bed shear stress in the river
channel is reduced at upstream of the bridge, that causing significant sediment deposition here.

In terms of sediment deposition in the river channel, though the post-flood bed elevation is shown in Figure 15, it should be noted that these are not direct comparisons, as the lines in cases 1 to 3 show the calculated bed elevation at the moment of the peak flow. The figure shows that in case 3, a large amount of sediment has already been deposited in the channel prior to the peak flow, which significantly reduces the channel capacity prior to the peak flow. Such sediment deposition is also seen in
the observed post-flood elevation. In Case 2, the sediment deposition in the river channel is also significant; however, the amount of sediment deposition is not as large as in Case 3 because the deposition of large wood at the bridge and the associated flow obstructions are not calculated. Due to these effects, the flood inundation expands over the valley bottom, as is especially noticeable near the bridge in Case 3 in Figure 12.

Figure 16 shows the contours of the flow velocity in the vicinity of the bridge (1.2 to 1.5 km). Figure 16 compares Case 2, in
which large wood is not computed, with Case 3, in which large wood is computed. The flow is obstructed in the bridge due to
the large wood accumulation at the bridge, causing the flow to divert around it. This results in a larger area of inundation in
Case 3 and a larger area subject to higher velocity fields.

## 5 Discussions

### 5.1 Assessments of key model assumptions

The present paper proposes a new method capable of simulating the behavior of large numbers of large wood pieces in the
flow field based on the convection and the storage equation. Figure 17 compares the experiment and computation results in
terms of the number of log jams and the log entrapment rate. Most of the calculation results range within the experiment results,
which show the validity of the present method in terms of large wood erosion and deposition. In cases without log jams,
although the log entrapment rate ranges from 0 to 3% within a 10m long channel, if the channel distance is long, such as a
river channel, a lot of wood will be deposited along with the flow associated with sediment deposition. In terms of wood
deposition, function $r(t, x, y)$ is a major factor that influences the results, taking into account the diameter of the wood and
also the presence of branches; in cases where the function is in case 2, the log entrapment rate is 70% higher than in case 1. In
cases with log jams, the entrapment rate of large wood, $p_b$, should be further investigated.

In addition, in terms of the model application to the field, the accuracy of the bed deformation has a significant effect on the
results.  For example, Figure 14 compares the measured and calculated results for large wood deposition.  In area (b), where
large wood tends to accumulate near the bridge, the observed and calculated results are in some agreement. On the other hand,
in area (a), the observed results show that large wood is deposited far from the original river channel, while the calculated
results show that large wood is deposited close to the original river channel, i.e., the right side of the white dotted rectangle.
In this area, the flow that is separated from the main flow becomes an eddy and deposits suspended sediment and large wood
in some places far from the original river channel, while the phenomena are not well reproduced in the computation due to the
grid scale problem. This means that in order to accurately evaluate large wood deposition within a 2-D flow scale, a fine mesh
must be used to evaluate sediment deposition.

### 5.2 Comparisons to previous modelling attempts

Previous Lagrangian modeling attempts, such as those proposed by Shimizu et al. (2006) and Ruiz-Villanueva et al. (2014),
can accurately analyze the behavior of large wood by tracking individual pieces of wood. On the other hand, in order to evaluate
the behavior of large wood in disasters such as the Akatani river, where as many as 19,500 pieces of large wood are produced,
it is difficult to use such existing Laglangian methods, while the present method can overcome such difficulties. The present
method also allows for basin scale analysis of the production, transport, and deposition processes of large wood, which is

useful even when the basin covers a large area. In a 2-D flow model, large wood behaviour is easily treated even in a field with significant river bed deformations.

On the other hand, as already described, the disadvantage of this approach is that the conversion from large wood concentration to actual large wood pieces and vice versa, such as wood deposition from water to bed, wood recruitment from bed to water, and large wood entrapment on structures, is not necessarily adequate at this stage. These points can be improved in the future by performing hydraulic experiments or, for example, by using the method of Kimura et al. (2021), which allows accurate calculations of the behavior of large wood in a flow field with wood branches. Another disadvantage is, since the large wood erosion and deposition is proportional to the sediment erosion and deposition, the precision of the bed deformation calculation directly affects the results of large wood erosion and deposition. In this sense, in order to obtain a reasonable result, it is necessary to use a sufficiently fine mesh when computing a 2-D flow model that can reproduce, for example, an eddy separated from a main flow.

## 6 Conclusion

The present paper has proposed a method to evaluate the behavior of large wood in the flow field based on the convection equation and the storage equation with active sediment transportation and channel bed deformation, which characterizes recent flood disasters in mountainous and hilly regions, such as the flood disaster in the Akatani river in 2017. Within the method, to investigate the validity of the erosion and deposition terms of large wood, flume experiments results are compared with the calculation results. The experimental results show that large wood is trapped in their sediment deposition areas, which are also reproduced by numerical simulations. In terms of the wood deposition, the function $r(t, x, y)$ is a major factor that influences the results, varying it within realistic values resulted in a maximum difference of about 70% in the rate of large wood deposition, and these results are still in the range of the experimental results. In cases with the log jam formed, though the large wood capture rate $p_b$ should be further investigated, the large wood capture rates are within the range of the experimental results, indicating the validity of the proposed method. The results of 2-D flood flow calculations with sediment and large wood in the Akatani river flood disaster suggested that large wood deposition is reproduced where bed deformation is well reproduced. However, there are areas in the computational results where the flow pattern and subsequent bed deformation are not properly reproduced, the large wood deposition is not well reproduced.

Since the proposed method makes it possible to simulate the behavior of a large number of large wood pieces, it can be applied to the management of hazards, such as the Akatani river. The computed results are useful for obtaining the effectiveness of countermeasures, developing hazard maps, and evacuation plans. In addition, the effect of countermeasures such as large wood capturing structures can be evaluated through simulations using the proposed method, which provides practical information to control hazards more efficiently and effectively.

## Data availability

The data used in this study are freely available  from the corresponding author upon request.

## Author contributions

DH and SE designed the study. DH performed the numerical simulations, and wrote the paper. SE reviewed and edited the paper.

## Competing interests

The authors declare that they have no conflict of interest.

## Acknowledgements

The authors would like to thank Dr. Nagumo, N., Mr. Nakamura, Y. and Dr. Yamazaki, Y. for their contribution to field survey and data preparation. We also thank Dr. Takahiro ITOH, El KOEI Co., Ltd. for his information for the experimental results.

## Financial support

This work was supported by JSPS KAKENHI Grant Number 22K14334.

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

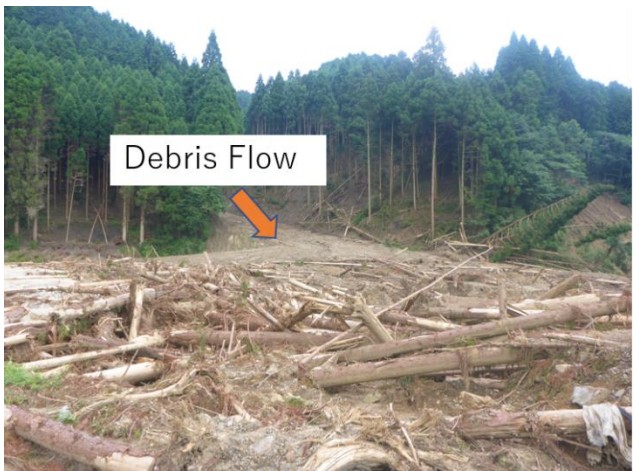 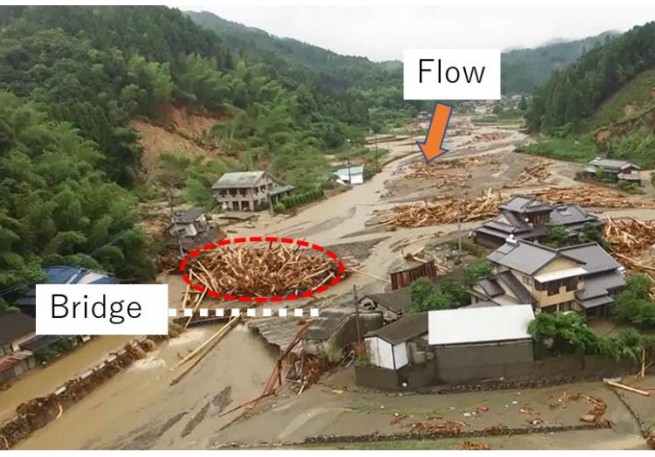

**Figure 1: Large wood deposition at the outlet of a valley bottom (left) and large wood depositions at around the bridges (right) in the Akatani river flood disaster, 2017. The left photo was taken by the authors and the right photo was taken by the Geospatial Information Authority of Japan.**

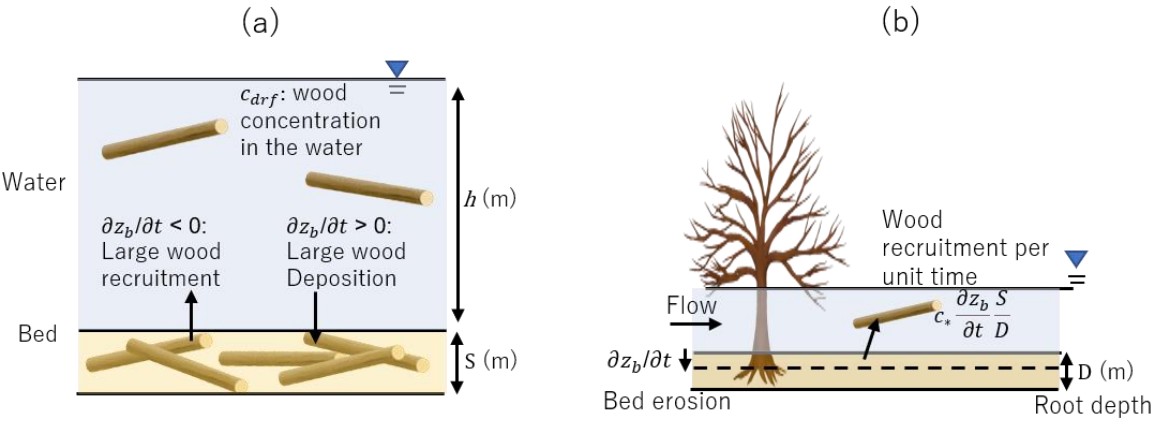

**Figure 2: Concept of large wood recruitment and deposition (a), and the relation between bed erosion, root depth and large wood recruitment (b).**

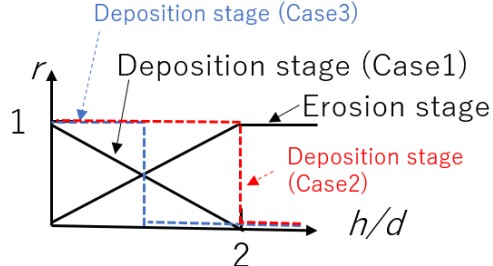

**Figure 3: Specification of the functional form of $r(t, x, y)$**

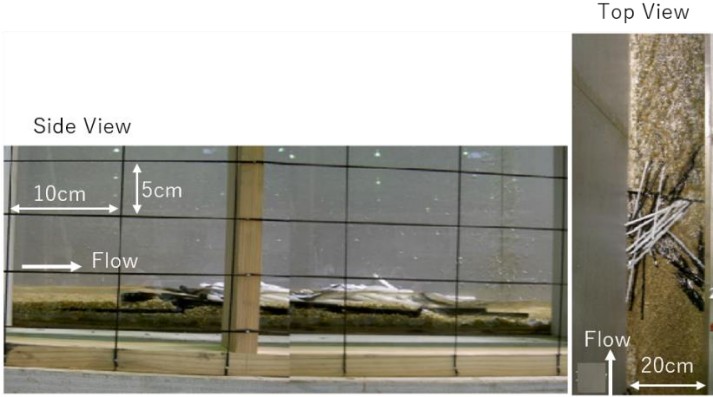

**Figure 4: Log jam formation in run 15; the authors modified the Itoh et al. (2010) photos.**

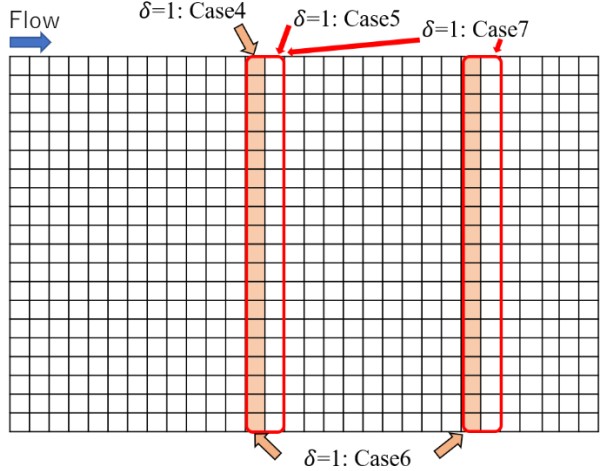

**Figure 5: Obstacle allocation concept in the computational domain**

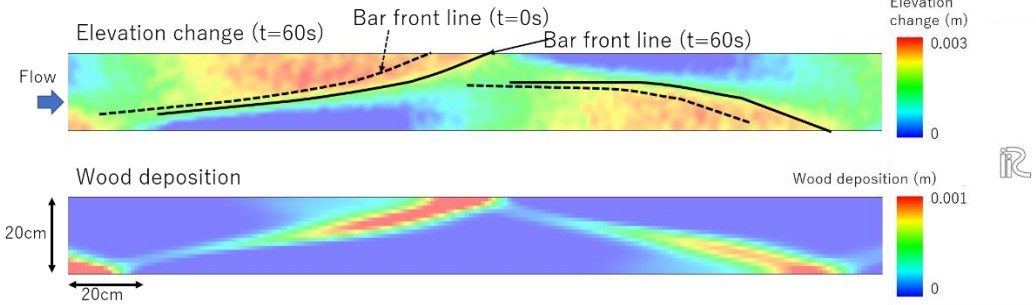

 **Figure 6: Relation between the computed elevation change (upper) and wood deposition (lower) in case 1. t=0s is the time just before the wood supply begins, and the solid line shows the front line of the sand bar after 60 seconds, meaning that sediment deposition occurs between these two lines due to the sand bar migration, and the wood deposition occurs where sediment deposition occurs.**

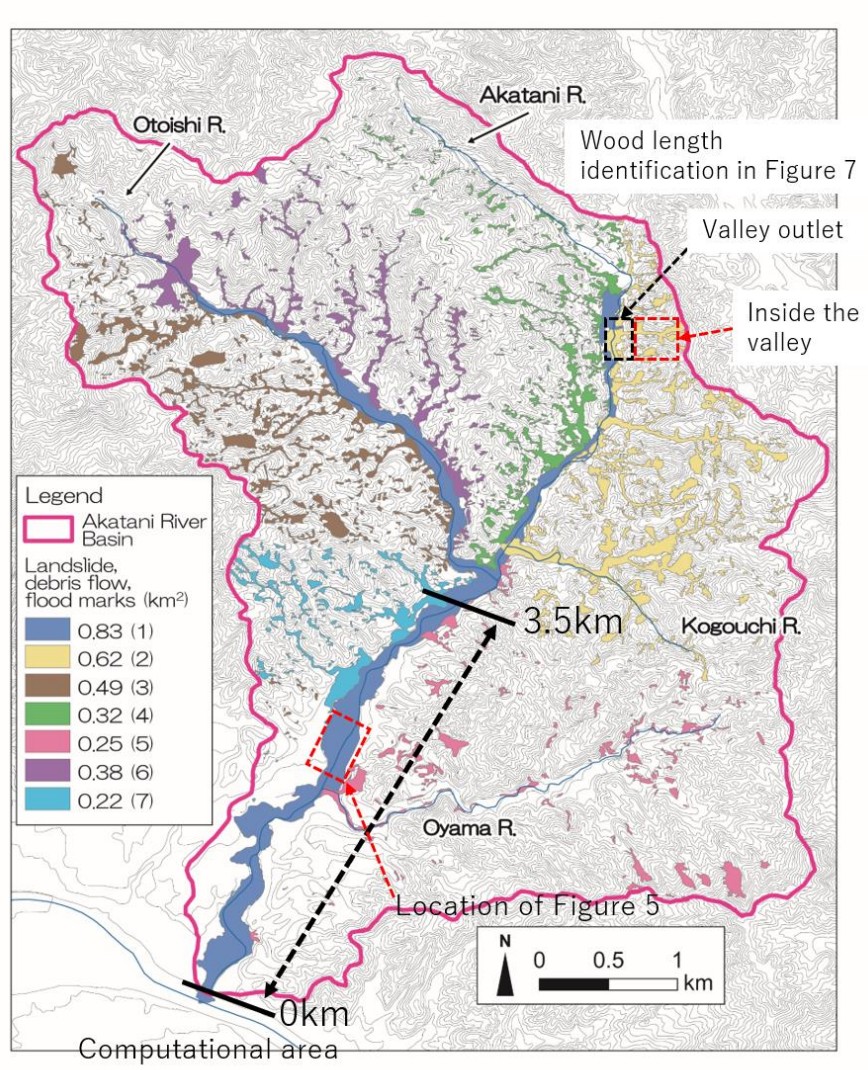

 **Figure 7: The Akatani River basin with debris flows and flood marks identified from aerial photos (Nagumo et al., 2019 was modified by the authors). The background image is provided by the Geographical Information Authority of Japan. The debris flows and flood marks are color-coded to identify the tendency of sediment supply: (1) is along the channels, (2) is the left bank side of the Akatani river basin, (3) is the right bank side of the Otoishi river basin, (4) is the right bank side of the Akatani river basin, (5) is the Oyama river basin, (6) is the left bank side of the Otoishi river basin, and (7) is the right bank side of the Akatani river basin.**

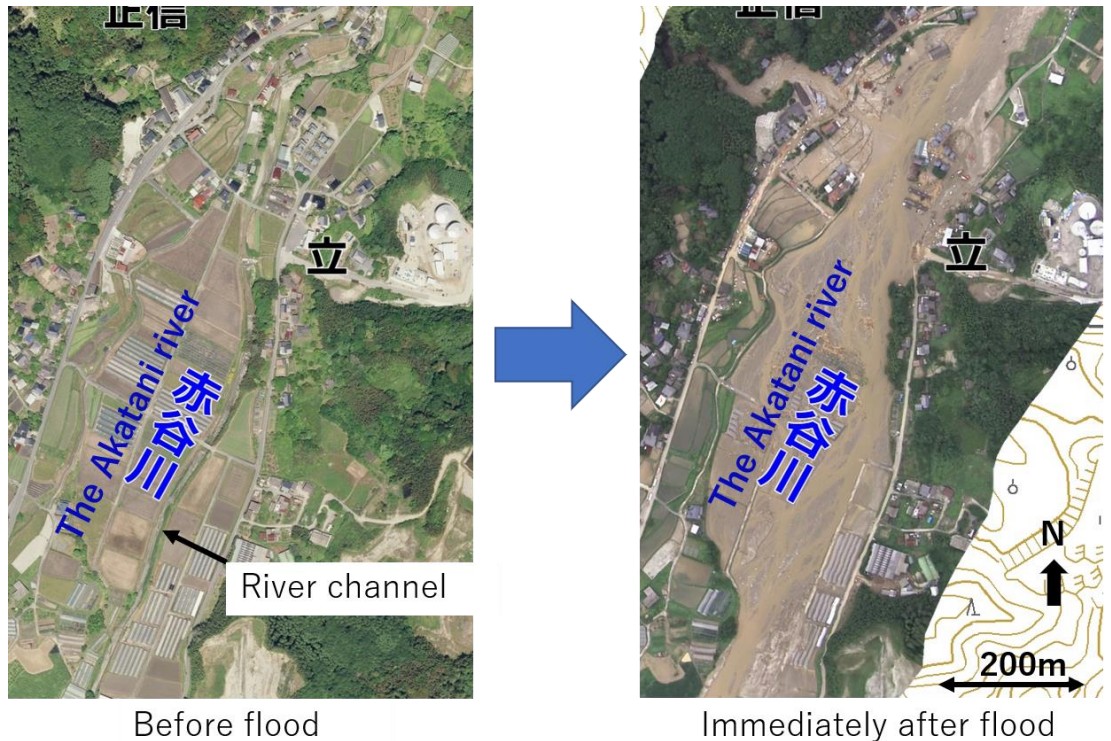

**Figure 8: Aerial photos of the Akatani River before (left) and after (right) the flood event in July 2017. The background image provided by the Geographical Information Authority of Japan.**

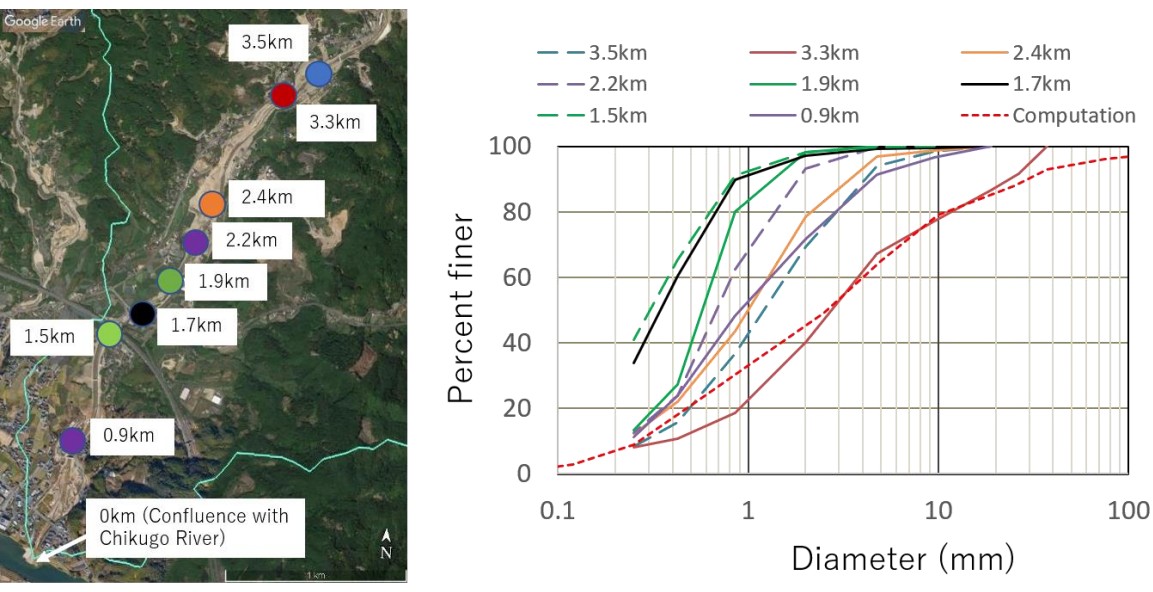

**Figure 9: Sediment sampling sites (left photo) and the sediment size distribution observed immediately after the flood event (right figure). The longitudinal distance corresponds to that of Figure 4. The green line in the left figure shows the basin boundary. The red dotted line in the right figure is employed as the initial condition in the computation. The background image was taken from © Google Maps.**

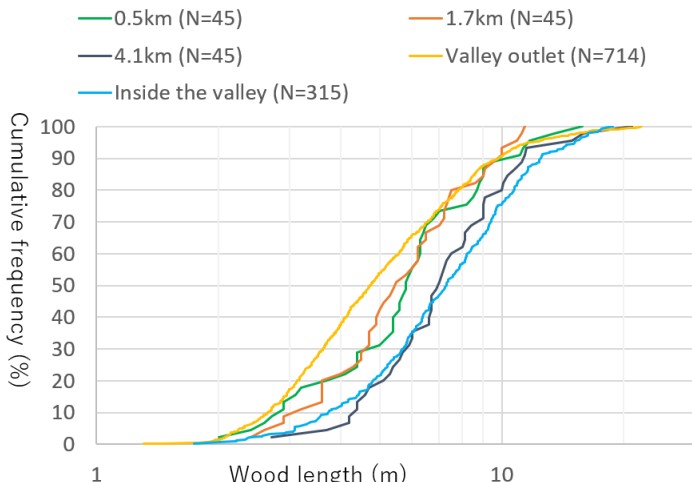

**Figure 10: Distribution of wood length identified from aerial photos taken just after the event. Location of the valley is shown in Figure 4. The identified wood length is shown as cumulative curves.**

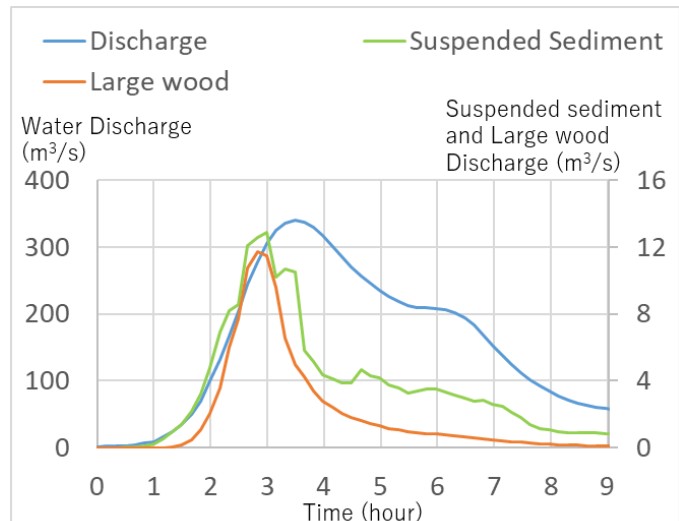

**Figure 11: Computed results, i.e., upstream boundary conditions for 2-D computation, for flood water, suspended sediment, and large wood discharge at the 3.5 km point.**


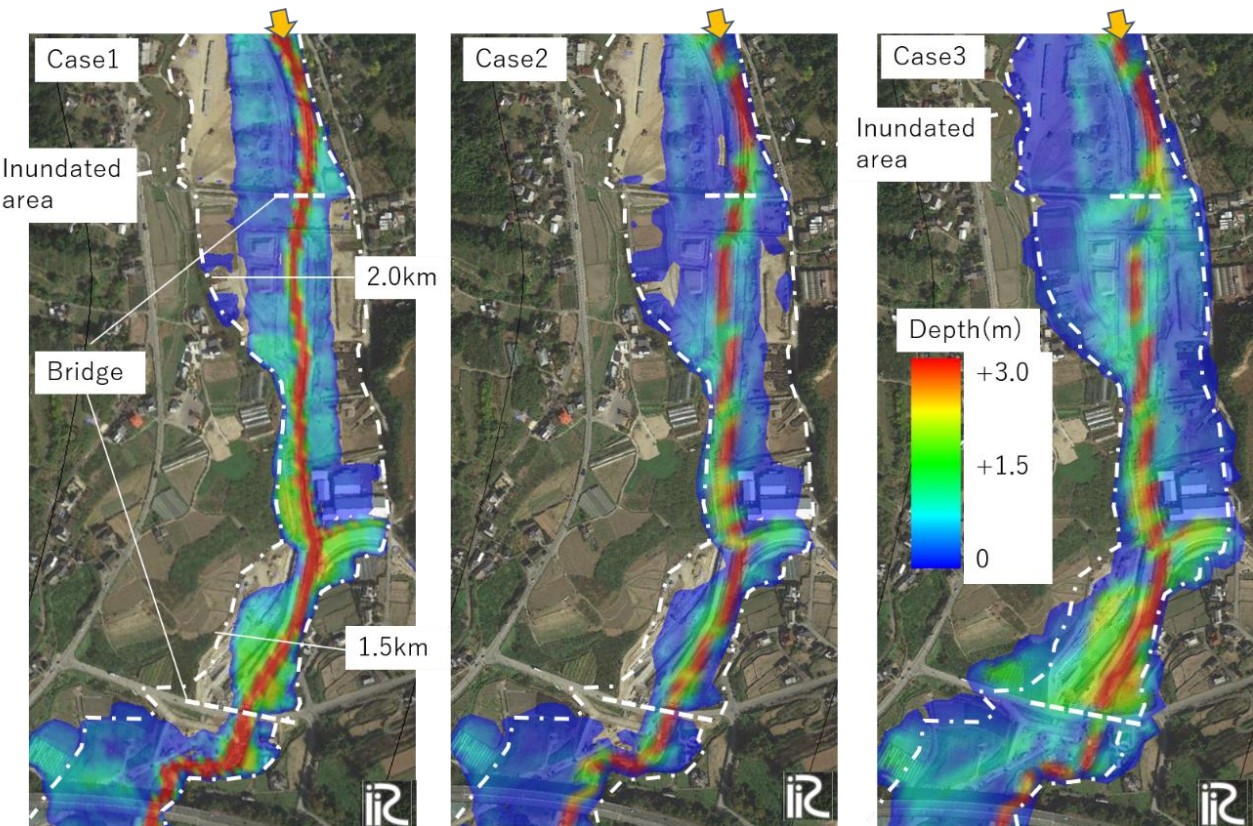

**Figure 12: Comparison of water depth at peak discharge between Case 1 (left) , Case 2 (middle), and Case 3 (right). Case 1 is the flow only, Case 2 is flow with sediment without large wood, and Case 3 is flow with sediment and large wood. The white dotted line indicates the inundated area as deciphered from aerial photo. The background image was taken from © Google Maps.**


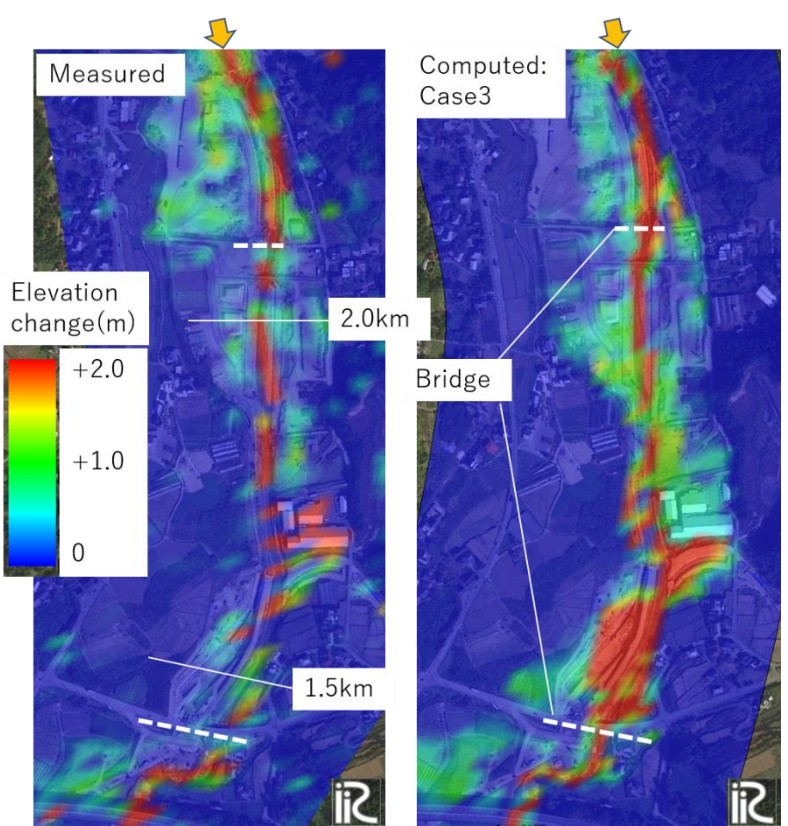

**Figure 13: Comparison of elevation changes before and after the flooding measured by aerial laser survey (left) and elevation change at the end of Case 3 calculation (right). The background image was taken from © Google Maps.**

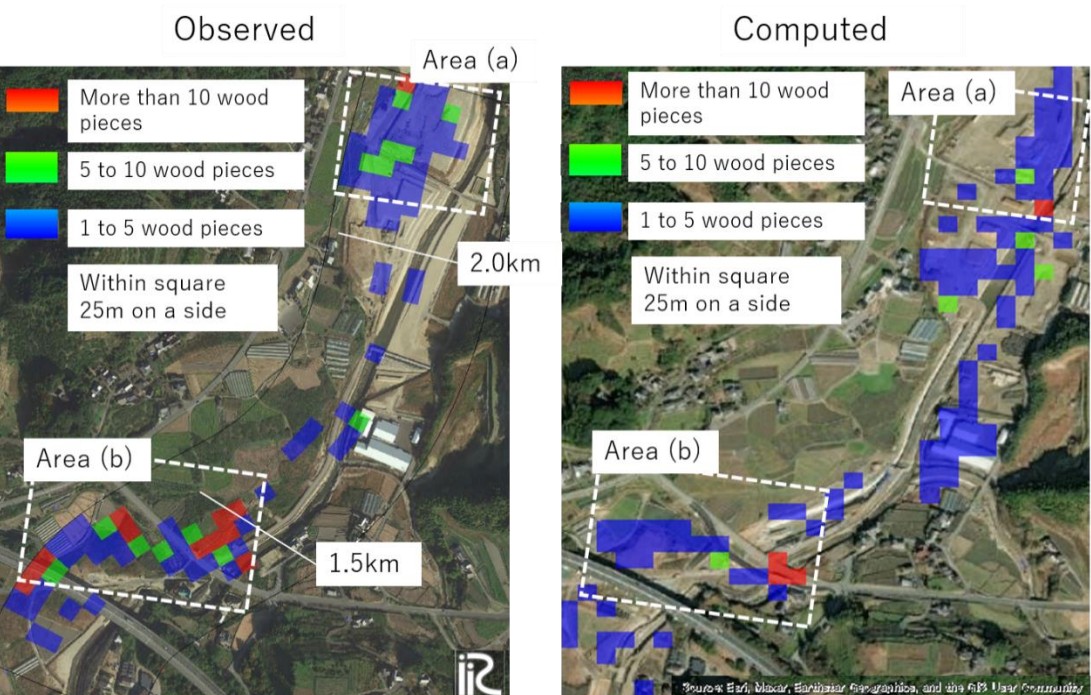

**Figure 14: Comparison of the large wood deposition between the observed from the aerial photos (left) and computed results (right). The computed results are converted to the number of pieces of large wood pieces by assuming that the diameter and length of a piece of wood are 20 cm and 7 m, respectively. The background image was taken from © Google Maps.**

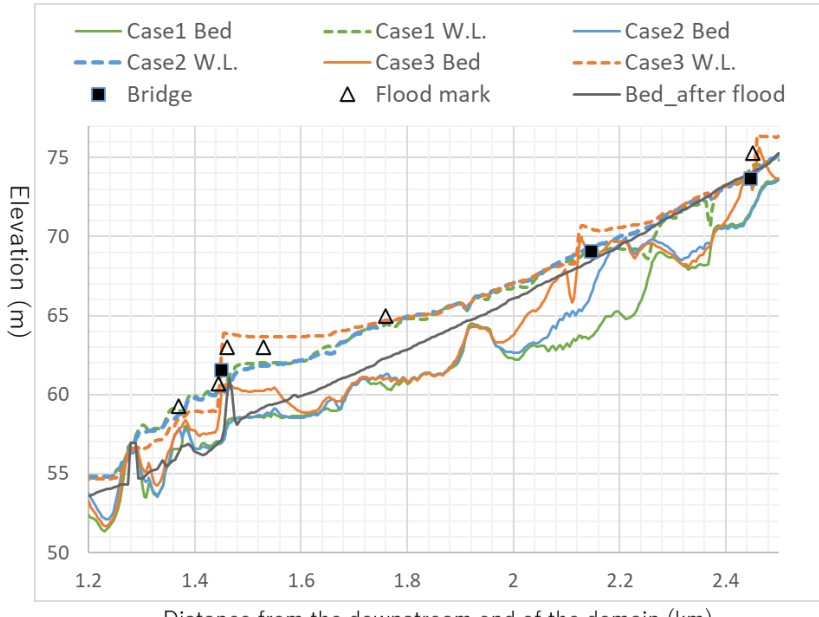

**Figure 15: Comparison of the results of each case with the water level mark and river bed elevation in the longitudinal direction of the river channel during peak flow. Bridge location ■ shows the elevation of the bridge height.**

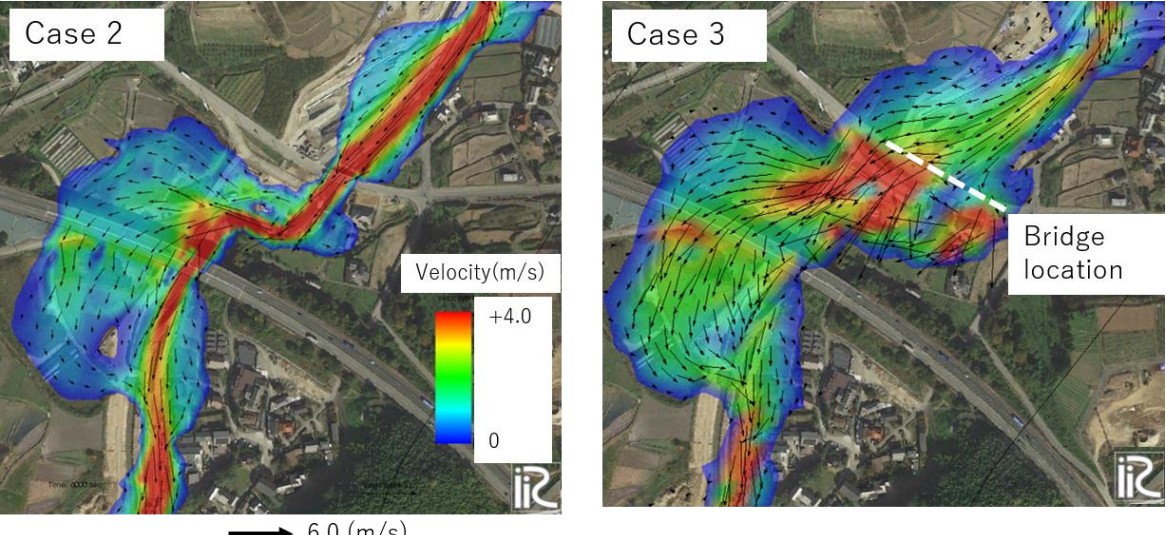

**Figure 16: Difference in flow pattern between Case 2 (left) and Case 3 (right) around the bridge at peak discharge. The background image was taken from © Google Maps.**

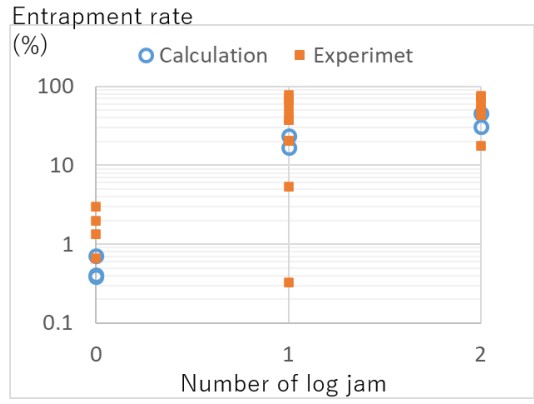


**Figure 17: Comparison of the flume experiment and calculation in terms of the number of log jam and the log entrapment rate. The original data is shown in Tables 1 and 2.**

**Table 1. Experiment cases and results conducted by Itoh et al. (2010)**

| Experiment Case | Supplied Logs(/s) | Specific weight | Num of Log jam | Trap rate (%) |
|---|---|---|---|---|
| Run9 | 1 | 0.952 | 0 | 0.0 |
| Run10_a | 6 | 0.952 | 1 | 0.3 |
| Run10_b | 6 | 0.952 | 1 | 48.0 |
| Run10_c | 6 | 0.952 | 0 | 3.0 |
| Run11_a | 10 | 0.952 | 2 | 58.0 |
| Run11_b | 10 | 0.952 | 2 | 76.0 |
| Run11_c | 10 | 0.952 | 2 | 72.0 |
| Run12_a | 1 | 1.2 | 0 | 0.0 |
| Run12_b | 1 | 1.2 | 1 | 62.0 |
| Run12_c | 1 | 1.2 | 1 | 38.0 |
| Run13_a | 6 | 1.2 | 0 | 0.0 |
| Run13_b | 6 | 1.2 | 0 | 0.7 |
| Run13_c | 6 | 1.2 | 1 | 68.3 |
| Run14_a | 10 | 1.2 | 1 | 77.2 |
| Run14_b | 10 | 1.2 | 1 | 5.4 |
| Run14_c | 10 | 1.2 | 0 | 2.0 |
| Run15_a | 6 | Mix | 2 | 17.7 |
| Run15_b | 6 | Mix | 0 | 1.3 |
| Run15_c | 6 | Mix | 1 | 20.3 |
| Run16_a | 10 | Mix | 2 | 51.8 |
| Run16_b | 10 | Mix | 2 | 43.0 |
| Run16_c | 10 | Mix | 1 | 20.4 |

**545   Table 2. Calculation cases and results for the entrapment rates of wood in the channel**

| Case | Function r(t,x,y) | Num of Log jam | Trap rate (%) |
|---|---|---|---|
| Case1 | 1 | 0 | 0.41 |
| Case2 | 2 | 0 | 0.71 |
| Case3 | 3 | 0 | 0.39 |
| Case4 | 1 | 1 | 16.7 |
| Case5 | 1 | 1 (2 lines) | 23.4 |
| Case6 | 1 | 2 | 31.2 |
| Case7 | 1 | 2 (2 lines) | 45.9 |

**Table 3. Parameters employed for the rainfall-runoff and landslide computations.**

| Item | Value |
|---|---|
| Mesh size (m) | 10×10 |
| Soil depth (m) | 1.0 |

| Saturated hydraulic conductivity (cm/s) | 0.5 |
|---|---|
| Equivalent roughness coefficient | 0.4 |
| Soil porosity: $\lambda$ | 0.475 |
| Internal friction angle (degrees) | 35 |
| Cohesion (kN/m$^2$) | 12.5 |
| Sediment density (kg/m$^3$) | 2650 |
| Water density (kg/m$^3$) | 1000 |

**Table 4. Calculation conditions for the 2-D flood flow with sediment and large wood**

| Case 1 | Flow only |
|---|---|
| Case 2 | Flow with sediment, without large wood |
| Case 3 | Flow with sediment and large wood |
