# Peer review of "Method to evaluate large wood behavior in terms of convection equation associated with sediment erosion and deposition"

_Earth Surface Dynamics, 2022_

## Author Response (AR1)

A point-by-point response to the reviews including a list of all relevant changes made in the manuscript

Comment from Anonymous Referee #1
This study attempts to model a very challenging problem of wood accumulation and flood hazard in extreme events , using a devastating 2017 event in Japan as the case study. The model builds on some previous work to define the boundary conditions and describe the quantities and size distributions of the sediment and wood accumulations that occurred during this event. As such I think it is an important attempt to offer a practical model for dealing with highly uncertain but incredibly damaging events that exceed the design conditions for which bridges and channels were designed. Ultimately the effort falls short in a few different areas in my opinion and the assumptions should be better justified and tested as part of this effort.

Answer to Anonymous Referee #1
We really appreciate the time and effort you have dedicated to providing insightful feedback on ways to strengthen our paper. In this revision, we tried to explain all the unclear points in this paper that you have pointed out. Below is a point-by-point response.

11 – 'aims to propose' could be just 'proposes'
Answer; We revised the manuscript.

21 – not clear to me in the abstract what type of model you are proposing. Is it a wood budget model or is it a flood level (hydraulic) model, or both?
Answer; We revised the manuscript to make the point clearer. The following sentence was added; 'This study proposes a new method to simulate the behavior of numerous large wood pieces in the flow field with active sediment transportation by employing the convection equation and the storage equation for large wood.'

27 – should use past tense to describe 2017 event.
Answer; We modified all the relevant descriptions in Section 1.

48 – I would clarify that these are modelling Lagrangian motion rather than tracking. There has been other research attempting to physically track the motion of wood, but that is not what is being reviewed, so you should be clear that this is limited to numerical models. Note as well that there are other wood budgeting efforts that don't use Lagrangian approaches, but

do look at trapping efficiencies and the like. I'm thinking of the Benda et al papers on stochastic forcing of wood budgets in watersheds. It would be good to see this new contribution placed in that sort of context in the literature review.

Answer; Thank you very much for the suggestion. We added Benda et al papers as a reference, and tried to clarify the author's points in the manuscript.

52 – do you have an estimate for how many? At what point is the Lagrangian method infeasible?

Answer; We added the estimation regarding the large wood supply in this event in Section 1. In addition, we added Figure 1 to describe the situation of the disaster.

93 – is that a good assumption? I have no information other than what I can imagine such as wood and sediment accumulating in deltas or alluvial fan locations, but what about wood moving into the floodplain and racking on floodplain trees while the sediment is more or less in equilibrium? I would like to see a bit of a review on how sediment deposits in watersheds. Wood can travel long distances in floods, much longer than sediment. How do you resolve the different virtual velocities of these two components?

Answer;

This concept is based on the situation that considerable amount of sediment and large wood were deposited where the sediment transport capacity was suddenly decreased in the flow direction in places such as fan topography formed by debris flow, as shown in the left photo in Figure 1; This explanation was added in section 1. In addition, to show the flood disaster and wood deposition clearer, we added Figure 1.

118- I wonder if this function could be applied to more than just bridges.

Answer;

We modified relevant descriptions; such as 'structures such as bridges'.

147 – figure 6 is not effective. I have a hard time understanding how to see the lateral distribution effect on Figure 6.

Answer;

We are sorry for the poor figure in the original manuscript. We improved the figure and captions (New Figure 7). Location of the valley is added in Figure 4.

159 – sentence beginning with "As for the water discharge…" is not clear. The authors need to explain in more detail. Clarify what is methods and what the results. A discussion point

of comparing with other vaguely cited reports is also presented very quickly. What exactly is presented in other reports. And what is the discharge?

Answer;

We described this part in more detail. Since there is no hydrological record in the Akatani River basin, the model parameters were validated using the Kagetsu river basin data, which locates east of the study basin.

175 – the section on boundary conditions is critical for the success of the model but is presented very quickly, citing mainly another paper Harada et al 2022 (Entrainment of bed sediment composed of very fine material in ESPL). It is possible that the authors are citing Harada and Egashira 2022, which has a more relevant title (Methods to analyse flood flow with a huge amount of sediment and driftwood), but this publication is in JSCE in Japanese and so will not be accessible to most readers. I think more detail is needed on these upstream boundary conditions given that they are models in themselves rather than measured time series. The authors tend to say things like "The occurrence of landslide, debris flow, and large wood transport induced by the landslide on the hill slope are evaluated based on the method of Yamazaki et al. (2019)," which seems to assume that we are all familiar with these methods. I at least am not and will need more detail on the basic steps, even if further details may be in the 2019 citation. In another sentence the authors say "in which a section that includes the upstream confluence and excludes the downstream confluence point is designated as the unit channel, and the sediment and large wood runoff for the entire basin is predicted by allocating the unit channels in series and parallel." I'm sorry but this is too vague.

Answer;

We described this part (section 3.2) more detail so that the readers could understand the outline of the process to estimate the time series of water, sediment, and large wood discharged from the basin. Indeed, we understand that more detail descriptions and discussions, such as sensitivity analysis or statistical discussions might be required, since the purpose of this study is to describe the 'method to evaluate large wood behavior in terms of convection equation', we would like to explain the process here only to the extent that allows the reader to understand the process.

182 – a lot of calibration in hydraulics is done with the roughness coefficient alone. It would be worth commenting on whether you could calibrate your model with just n. Is 0.03 justified or is it just because it is a typical value?

Answer;

The roughness coefficient is set as 0.03 for the entire computation domain, which was

determined so that the flood marks would match the computation results. This sentence was added in the manuscript.

180 – earlier you said the average slope was 1/70. Which is it or over what reaches do these averages apply?

Answer;

The bed slope in section 3.1 is the bed slope for the entire Akatani River channel (1/70), and the slope in section 3.3 is bed slope of the computational domain within the 3.5km (1/120). We clarified these points.

191 – I might have missed this, but where are the cases described?    I see some description in the Discussion (lines 215), but this should have been clearly presented in the methods.

Answer;

The case descriptions were added in the manuscript.

230 – This sentence seems to be the justification for assuming that wood and sediment behave the same in terms of erosion and deposition. I found this assumption questionable given the different properties of the wood and sediment, with the key difference being the density relative to water.   Wood travels much faster and at the water surface, interacting more with the banks rather than the sedimentary bedforms, at least at high flow conditions. Figure 10, for example shows that wood deposits far from the main channels, with only minor correlation with the bridges.   I don't think that the correlation between the observed and computed figures is strong enough to say that the assumption is correct.    The wood tends to accumulate far from the thalweg, but the bridges are the highest modelled accumulation points because of the equations used and assuming values of 1.0 with respect to the amount of wood trapped.

Answer;

With regard to the reviewer's point; "Wood travels much faster and at the water surface, interacting more with the banks rather than the sedimentary bedforms, at least at high flow conditions. " In our model, as shown in Equations (14)-(17) and Figure 3, if the ratio of water depth and diameter of a wood piece exceeds 2, large wood is not deposited on the riverbed. In other words, if the water depth is deep enough, large wood pieces are not accumulated on the riverbed and flows downstream with the flow.

The difference between the left and right sides of Figure-11, i.e., the correspondence between the observed and calculated results of driftwood, has been added in the manuscript as follows; With regard to the spatial distribution of large wood deposition, the difference between the left and right figures of Figure 11 shows the correspondence between the observation and

calculation results of large wood. In area (b), where large wood tends to accumulate near the bridge, the observed and calculated results correspond to some extent. On the other hand, in area (a), the observation results show that large wood is deposited far from the original river channel, while the calculation results show that large wood is deposited close to the original river channel, i.e., the right side of the white dotted recutanglar. In this area, the flow that is separated from the main flow becomes an eddy and deposits suspended sediment and large wood at far from the original river channel, whereas the phenomena is not well repdoduced in the computation due to the issue of grid scale. Since the present method assume the large wood deposition occurs where sediment deposits as described in equations (14) and (15), the reproductivity of bed deformation greatly influences the results of large wood deposition.

Figure 5 – repetition of colors and different units are not clear at all. What does 3.5km-2 mean vs 3.5 km-1? Are these repetitions at the same location? Why are two needed? The color scheme for the size distribution is not helpful and overall the figure quality is poor – location diagram is a google earth photo with pins dropped at the approximate locations.
Answer; We reduced the number of lines and made both figures much clearer. (New Figure 6)

Figure 6 – wood length distribution legend not clear. What areas are used? What does inside the valley mean? Can these be put on a map somewhere?
Answer; We improved the figure and captions (New Figure 7). Location of the valley is added in Figure 4.

Figure 8 – best to reiterate what the difference is between cases 1, 2 and 3 in caption
Answer; We added the cases in cation. (New Figure 9)

Figure 10 – legend should say 'wood pieces' instead of 'woods'
Answer; The legend was improved. (New Figure 11)

Figure 11 – figure quality poor due to positions of the axis labels and the low resolution of the graphic. Hopefully vector versions of these plots will be provided.
Answer; We improved the figure qualities. (New Figure 12)

Dear authors,

I have read your ms "Method to evaluate large wood behavior in terms of convection equation associated with sediment erosion and deposition" with great interest. The prediction of wood dynamics (the term "driftwood" should be used for wood elements floating in lakes and oceans only, not in rivers) during large floods is undoubtedly crucial in many regions worldwide, and available models are still few and very often untested against real events.

Having said this, I do not think the model proposed in the ms represents a step forward in our ability to predict wood transport, and thus manage potential wood hazards. Although the effort made by the authors in trying to express mathematically wood transport processes in Eulerian terms is understandable, it is well known on one hand that a model should simplify the reality only up to the point that dominant driving factors are still captured. On the other hand, a model should not be more complex than needed for its purposes. While the authors claim that the model has been successfully validated against the 2017 post-flood surveys, in my opinion this is not the case. The major points of criticism from my side are reported below, and I hope they may help you.

Best wishes

The 2D model is claimed in the introduction to be able to "describe the behavior of large wood based on the convection equation and the storage equation with sediment erosion and deposition to simulate the behavior of numerous numbers of large wood pieces".   If I am not wrong, the present formulation – for both sediment and wood – neglects entirely the bank erosion process. Such process is well known to be the dominant wood supply mechanism in partly-confined and unconfined rivers, and bedload rates may also be greatly influenced by lateral channel migration.   Also in the case of the Akatani river bank erosion/channel widening seems (based on Fig. 4) to have been a massive "player" during the flood;

The model to compute the "amount of sediment and large wood inflow from the basin at the upstream boundary of the 2-D analysis" is swiftly presented, with too few details and insights about its plausibility/performance. Uncertainties in the prediction of mass movements processes (location, volumes, connectivity with the channel network) coupled with their wood supply are huge, and there is not track of this in how the upstream boundary conditions have

been later used for the 2D simulations. Furthermore, the forest stand parameters are said to have been "assumed", but in such a relatively large basin area a constant value for them is highly unlikely to be real. Several different wood input scenarios should have been tested at least, integrated with bedload transport scenarios;

The validation of the model is absolutely not convincing, being proposed in a highly qualitative and non-systematic way though the domain. In addition, the arguments brought to support that idea that the model has been successful mostly rely on comparing – again too vaguely, in semi-quantitative terms at best – the flow field and on the deposition pattern in the proximity of the bridges. But this is an "easy win", as for sure bridges are areas where wood was trapped (Dirac delta imposed = 1) and thus flow (increase in flow depth and diversion around the bridge) and sediment transport (deposition) were affected. Therefore, I would say that the model validation suffers from both strong equifinality issues due to its large number of unconstrained parameters and from a tautological argumentation without an accurate, statistically-based accuracy analysis, recalling also the uncertainties regarding the input conditions.

Regarding the practical outcomes of the model proposed, the same conclusions about the role of bridges in the Akatani flood could have been obtained by applying any hydraulic or morphodynamic model with the use of reduced cross-sectional areas at bridges due to expected high wood load. As the authors correctly say, an accurate estimation of the extent to which wood may clog a bridge is crucial for this aim, but this is not incorporated in the proposed model. I was a river manager, I'd certainly use simple, empirically-based rules to include the role of wood and sediment deposition at bridges in robust hydraulic models rather than using a very complex, time-consuming 2D model whose outputs are subject to large epistemic and aleatory uncertainty.

Answer to Anonymous Referee #2

We really appreciate the time and effort you have dedicated to providing insightful feedback for our paper. After perusing the reviewer's remarks, we found that there are several parts in which our intentions have not been communicated. For example, what type of disaster it was and that we are conducting this study to evaluate large wood behavior for such a disaster. In other words, we think your point that "a model should not be more complex than needed for its purposes" is a criticism because the 'needs' and 'purpose' are not well conveyed.

Therefore, in this revision, we have first made significant additions to Section 1 to provide a more detailed description of the disaster. For example, in the Akatani River disaster, it is estimated that approximately 19,500 pieces of large wood were produced by landslide and

debris flows, and that large amounts of sediment and large wood were supplied to the river. Thus, an appropriate evaluation of such large wood production, transport, and deposition process is the most fundamental aspect of numerical analysis for this event. The channel winding in Fig.5 took place due to the sediment deposition in the valley bottom, that is also clear from Fig.10, thus we hope the reviewer understands that bank erosion process is not a major factor that the authors intend to discuss.

We also believe that the bank erosion is one of main causes for large wood pieces. Equation (4) normally evaluates bank erosion in which x- and y- components of bed-load rate ($q_{bix}$, $q_{biy}$) are evaluated by the bed shear stress and the velocity in the vicinity of the bed where the secondary currents are produced due to curvature of stream lines, and thus the source of large wood is evaluated by Eq. (16) and (17) in case erosion occurs. Thank you for your valuable comments in this regard.

In response to the comment about the lack of explanation of the upstream end boundary conditions, we have made significant additions to Chapter 3.2. Indeed, we would like to write so much in chapter 3.2 that it should be a stand-alone paper, but that would make the paper too large. Therefore, we revised the chapter so that it makes some sense only in this paper. Regarding the point, 'Several different wood input scenarios should have been tested at least, integrated with bedload transport scenarios', please understand that we have performed parameter calibration to match the observed collapse area and large wood runoff estimation, as we have added in chapter 3.2.

We understand that your criticisms about the validation of the model, especially the lack of statistical discussion, that is what we need to pursue in the future. On the other hand, we hope you understand that it is not easy to obtain data on such disasters. For example, it is not easy to obtain data on the spatial distribution of large wood deposition after a disaster, because some large wood is buried under sediment. Indeed, purpose of the present paper is to show the concept of convection equation to analyze the large wood behavior and its applicability for the such extreme disaster. We understand that there are issues that need to be addressed in the future, including the large wood capture at the bridge (Dirac delta imposed = 1).

Finally, we appreciate your insightful comments again, and the revised sentences are marked in the revised manuscript.

---

## Author Response (AR2)

Author's response to the Associate Editor's comment

We really appreciate the time and effort you have dedicated to providing insightful feedback on ways to strengthen our paper. We have a full and thorough understanding of the points raised by the Associate Editor.

We added the section 3 that reproduces the hydraulic experiments to study the characteristics and test the validity of the proposed method. Though the option to move the entire field case study to a future publication was suggested, we remain the part in section 4. This is because our goal is to use the proposed method to assess, predict, and mitigate potential sediment and flood hazards in the filed rivers. Also, the results of the field study, such as Figure 14, raise the discussion for the importance of flow reproduction and associated sediment deposition in the 2-D model, so to obtain a reasonable result for large wood deposition in this model, it is necessary to use a sufficiently fine mesh when computing a 2-D flow model that can reproduce, for example, an eddy separated from a main flow.

Based on these calculations in Sections 3 and 4, Section 5, the discussion section, has also been completely rewritten, following the suggestions of (i) a critical assessment of the model's main assumptions, and (ii) a comparison with previous modeling attempts. Corresponding to the revisions in the Discussion section, the Introduction, or Section 1, clearly presents the position of these calculations, and the Conclusion, or Section 6, adds the conclusions from these additional investigations. The Abstract has also been modified to reflect these changes. We hope that this revision will address the points raised by the two reviewers and the Associate editor and move the manuscript forward for publication.

Best wishes,
Daisuke Harada and Shinji Egashira

Associate Editor's comment

thanks for the manuscript and your edits and rebuttal. I have looked through the reviews and your response and have decided to return the paper to you for some more revisions before sending it out for review again. Reviewer #2, in particular, has raised some serious concerns, which I do not think you have fully addressed yet. Reviewer #2 makes four major points, each of which seems justified and fair to me. The rebuttal, at the moment, presents some general arguments that only partially address the reviewer's concerns.

I do not think the concerns are fatal for the paper. Rather, my impression is that the specific objectives of the paper and of the comparison to the case study are not well communicated at the moment. I agree that the model description lacks some details and that it is unclear how it differs from / advances over previous attempts. Further, I agree with reviewer #2 that the data and the way you use it at the moment to not provide a convincing validation of the model. In general, the purpose of the field study is unclear. Yet, I also think that a validation is not entirely necessary for the paper to make a valuable contribution. Instead, you could treat the case study as an example application.

There are some other parts where I think the paper needs some development. In particular, the discussion is rather short at the moment.

So, here is what I suggest:
- clearly state at the end of the introduction as to what the objectives with the field data comparison are
- expand and sub-structure the discussion, to include sections (i) with a critical assessment of the model's main assumptions, and (ii) a comparison to previous modelling attempts. In the latter, please highlight where you see the advances, advantages and disadvantages of your formulation. I suggest to move the discussion of the case study into a separate sub-heading. You could also expand the last paragraph to include model requirements (what type of data is needed? What are limits in terms of catchment size and stream morphology?), and give some more information on the type of applications you envisage your model to be suitable for.
- for the field study, try to better work out the take-home messages for the reader. At the moment, the discussion merely contains a few statements on the water levels, the flow velocity, and the spatial distribution of wood. As a reader, I am not really sure what the purpose of these bits of information are and what I should learn about the model from it.
- it may help the readers and potential users to understand your model if you simulate and present some simpler instructive 'ideal' cases or numerical experiments in addition to the complex field case of the Akatani basin. For example, a straight river with a bridge, or a river bend, combined with a small parameter study varying wood load, peak discharge, etc. Given that you state in the rebuttal that concerning "the lack of statistical discussion, that is what we need to pursue in the future" - maybe it would be an option to move the entire field case study to a future publication, and focus in this paper on the model behavior using a number of suitable numerical experiments.

I hope this helps and I am looking forward to seeing your revised paper. Please get back to me

if you need further clarification.

With best wishes, Jens Turowski
Handling AE

Typos and notes
65 The basin had not experienced⋯
207 ⋯within the 3.5 km reach⋯
208 plesae give some more details (at least grid size)
209 ⋯is set to⋯
210 please include details of the calibration procedure
210 how was the sediment size distribution determined?
260 rectangle
262 reproduced
264 reproducability

---

## Author Response (AR3)

Author's response to the Associate Editor's comment

We really appreciate the time and effort you have dedicated to providing insightful feedback on ways to strengthen our paper. We have a full and thorough understanding of the points raised by the Associate Editor.

In this revision, we essentially reorganized and improved the chapter 3. Particularly, this time, we try to present the observations from the experiments in 3.2, then the results from the model in 3.3, and the comparison is moved to section 5.1. In addition, some discussions in section 5.2 are moved to chapter 1 to clarify the position of the present method, and the relationship to previous attempts from the literature. Also, we added a paragraph to start in general what we want to achieve in each section.

We hope that this revision will address the points raised by the two reviewers and the Associate editor and move the manuscript forward for publication.

Associate Editor's comment

I have now received two evaluations of the manuscript, from the original reviewers. Both laud your efforts in improving the paper, and suggest that although the paper got closer to the final version, some further work needs to be done. Reviewer #1 thinks the balance between field and lab work needs to be improved, and asks for further information in particular on the lab work. Reviewer #2 makes a similar comment, and requests some further clarifications on the new model, and the relationship to previous attempts from the literature.

On the latter point, I actually think some of the points you currently make in the 2nd chapter of the discussion present the motivation for the new model. In particular, I would suggest to move some of the text to the introduction. You can briefly introduce the lagrangian approach of previous models in the introduction, state (as is in the discussion at the moment) that this prohibits the simulation of large, complex events with many individual pieces, and that you are striving for a model solution closing this particular gap.

Generally, I think you can try to better separate methods, observations and interpretations, particularly in chapters 3 and 4.

In chapter 3, on the experiments, I think the major problem arises from an unclear distinction of model and experimental results. Here, I suggest to reorganize the writing, first presenting the observations from the experiments, then the results from the model. The comparison can be moved to the discussion. Further, I think it would help to precede the chapter with a paragraph stating in general what you want to achieve.

Similarly, you could start chapter 4 with a few sentences introducing the purpose of the comparison.
* * *
Comment from Referee #1

I appreciate the work that the authors have put in to better explaining their research and better justifying their statements. This revision goes a long way towards making the article understandable to a wide audience. Given the substantial revisions, I reread the article and commented on it in its entirety rather than focussing on my past comments.

This new version is much stronger, but some of the figures are still of lower quality and the explanations are not always clear. The addition of the laboratory experiments could be used to better explain the physical interpretation of the model results to contrast it with the Lagrangian models that are available. I think the work has value, but you need to better explain how wood is modelled as a depth. I would also put more emphasis on the result that the inundation limits and (possibly) the changes in channel bed elevation are best captured by including the wood effects in the model. From what I can see, the model without the wood misses some key details that are critical for predicting the flood hazard, so this is an important result.

I also recommend a review of the language to ensure that the meaning of the sentences are correct in English. The researchers have improved this part substantially, but there are still some points of confusion.

Author's response to Referee #1

We really appreciate the time and effort you have dedicated to providing insightful feedback on ways to strengthen our paper. As for the depth in the wood modeling, we totally reorganized the chapter 3 to investigate the erosion and deposition of large wood. As we emphasize in this revision, we assume two major factors that affect calculation results: the exchange of the large wood between the water and the bed and wood entrapment on structures. To investigate the validity of these factors, we compare the flume experiments and the calculations in chapter 3.

Also, as for the language, this time we asked an English proofreading specialist to review the manuscript to ensure that the meaning of the sentences are correct in English. We hope that this revision will address the points raised by the reviewers and move the manuscript forward for publication.

The answers to the specific comments are as follows;

32 – this description could be clearer. You are trying to separate the upstream and downstream processes, roughly divided into hillslope and debris flow type processes and flood flow and erosion of sediments and wood from the colluvial and fan deposits. I think you should be clearer to name the areas where deposits occur from the first set of processes and from the second set. I would like some idea of the size and steepness of the basins delivering sediment and wood to the main channel. I would also like some idea of what you consider to be the main channel. Could you use stream order to help differentiate? It is not that big of a basin, so I find the idea that the processes are so cleanly separated to be unlikely.

Answer:

We reorganized these points in the manuscript, using the idea of stream order.

75 – not sure if you mean 'field' rivers? I think you can just say 'rivers'.

Answer:

We modified it.

98 - maybe just mention that this determines the energy lost from solid particle collisions. What value is used for this?

Answer:

'e=0.85' was added in the manuscript.

113 - based on your response to the comment from previous version you should say that the buoyancy of the wood is considered with deposition. I know you later describe that the r value is also used, but I think it needs to be said here. The discussion of the shallow depths whoudl occur in this paragraph I think rather than at line 136

Answer:

We also added the explanation for the exchange from the neutral buoyant particles to the wood pieces at around the lines from 112 to 120.

136 – 'Meanwhile' not needed

Answer:

We removed it.

143 – what do you mean by the 'hidden' effect? Do you mean a feedback?

Answer:

We replaced 'hidden effect' by the 'shielding effect'

151 – vague statement that the 'applicability of the method is investigated···" Does this mean it is being validated or calibrated? Or is it more of a qualitative investigation of 'applicability'?
Answer:
We modified the sentence as follows: The validity of the proposed method is investigated by comparing the past flume experiments conducted by Itoh et al. (2010) and the calculations that reproduce the flume experiments.

197 - I don't follow the logic here. How are the flume experiments being used to validate the model? It seems that you are adding obstacle rows to observe the effect it has on log capture, but I don't see how this validates the model.
Answer:
We substantially modified this section 3, and the comparison between the experiment and the calculations are added in Section 5, the new Figure 17.

245 – here I don't think you mean validated. I think you mean estimated, or assumed based on the neighbouring Kagetsu basin. Validation is a procedure for testing statistical fits with a model after having calibrated the model.
Answer:
To be precise, the model parameters were calibrated using the Kagetsu river basin data, which locates east of the study basin, and the parameters are applied to estimate the flow discharge in the Akatani river basin. We added the sentence in the manuscript.

257 - It does help to show the boundary conditions over time as you have done in Figure 11. These are key to the model and difficult to constrain. At the end of the section it would be good to acknowledge the uncertainty of these predictions and state, as you did in your review, that these BCs are hard to validate but that they appear to be reasonable based on the information that you have. You could also emphasize that the model is sufficient for the overall purpose at this point, which is to model the spatial distribution of wood in extreme events where there is not much data and to include the effects of the wood in flood prediction, which is by far your best validation.
Answer:
We added the sentences at the end of chapter 4.2, following the important comment.

287 – the last sentence of the paragraph remains very broad that I find is not sufficiently

supported. How is 'generally consistent' defined? Is it just the visual comparison you are basing this on?

Answer:

We removed the 'very broad' sentence from this paragraph, and the comparison of this figure is moved to chapter 5.

297 – is there a post-flood scan of the bed that you could use to assess modelled bed elevations? The wood impact is clearly significant at a could of the bridges. Is this what was found in the field after this flood?

Answer:

We added the post-flood measured elevation, and added some relevant texts.

334 – 'can elaboratelyl evaluate' phrase is not clear. Maybe just 'detail' used as a verb?

Answer:

We modified the sentence as follows;

Previous attempts can accurately analyze the behavior of large wood by tracking individual pieces of wood.

338 – poor sentence. 'Thus this method must be effective for such cases' is not clear.

Answer:

We removed the part.

363 – what about the lab experiments? Why are they in there and how do they support your conclusions?

Answer:

We added these points in the manuscript.

Figure 4 - low quality figure. No scale in diagram, no indication of parts a and b. Presumably they are side view and top view, but this figure could be better.

Answer:

We improved the figure.

Figure 5 - more information needed in caption. Which run is this? I don't understand the units. Wood deposition calculated in mm should be given a physical interpretation.

Answer:

The new Figure 6 (Figure 5 and 6 are exchanged) is improved in the legends and captions.
* * *
Comment from Referee #2

Dear authors,

I think your ms has been ameliorated compared to the previous version, and now it is nearing a valuable contribution to the understanding of the challenges involved in wood transport modelling during floods.

However, I find the revised ms unbalanced between lab and field work, as the results from the lab work - which should convince the readers about the validity of the model applied to the Akatani River - are too poorly presented in my opinion. Figures on this part (4, 5, 6) are not that helpful in my view. Overall, I am not really sure this lab part helps the paper, to be honest. To achieve this goal, a more quantitative analysis of model's prediction performances against flume data should be presented, and possibly less figures (and text) should be dedicated to the field case.

A minor comment regards the parameters used in Tab 3: soil density is 2650 kg/m3, but I guess here you meant sediment density? because soil cannot be without porosity.

Best wishes

Author's response to Referee #2

We really appreciate the time and effort you have dedicated to providing insightful feedback on ways to strengthen our paper. As for the comment that the lab part does not help the paper, this time, we totally reorganized the chapter 3 to clearly state the model validation. We added Figure 17 for a visible comparison of the experiments and calculations. Also, texts in chapter 4 are reduced in this revision. The 'Soil density' in Table 3 was replaced by the 'sediment density'. We hope that this revision will address the points raised by the reviewers and move the manuscript forward for publication.

---

## Author Response (AR4)

Author's response to the Associate Editor's comment

We greatly appreciate the time and effort that the editor and reviewers put into providing insightful feedback on ways to strengthen our paper. We hereby submit our technical revision of the manuscript for further consideration.
Thank you very much again and best regards;
Daisuke HARADA

Comment from Referee #1
Dear colleagues, I think there is a mistake at line 121:
How can pore water contain large wood particles?! I guess you mean that the riverbed can contain wood, not pore water.
Best regards

Author's response to Referee #1
Since the previous sentence was not clear, we modified the sentence as follows;

This concept is consistent with the idea that when sediment is deposited on the riverbed, water is also deposited as pore water within the void of the sediment, and when sediment is eroded, pore water is also taken into the water, and this eroded sediment and pore water contains large wood, as the wood particles are treated in terms of wood concentration.

Comment from Referee #2
I have only two minor additional comments:
120 ' pore water' is deposited – What is pore water? Not sure if translation is correct here.
303 – should be plural "··· because they are buried."

Author's response to Referee #2
As for the line 120, we answered above.
We also modified line 303 as suggested.

Others;
We modified the figure captions for Figure 1 and 4 to show the photo's credits.